# A cardiac-null mutation of *Prdm16* causes hypotension in mice with cardiac hypertrophy via increased nitric oxide synthase 1

Ji-One Kang[1]⊙*, Tae Woong Ha[1], Hae-Un Jung[2], Ji Eun Lim[1], Bermseok Oh[1]⊙

1 Department of Biochemistry and Molecular Biology, School of Medicine, Kyung Hee University, Seoul, Korea, 2 Department of Biomedical Science, Graduate School, Kyung Hee University, Seoul, Korea

⊙ These authors contributed equally to this work.
* jionekang@yahoo.com

**Data Availability Statement:** All relevant data are within the paper and its Supporting Information files.

## Abstract

Hypertension or hypotension prevails as a comorbidity in patients with heart failure (HF). Although blood pressure (BP) is an important factor in managing the mortality of HF, the molecular mechanisms of changes in BP have not been clearly understood in cases of HF. We and others have demonstrated that a loss in PRDM16 causes hypertrophic cardiomyopathy, leading to HF. We aimed to determine whether BP is altered in mice that experience cardiac loss of *Prdm16* and identify the underlying mechanism of BP-associated changes. BP decreased significantly only in female mice with a cardiac-null mutation of *Prdm16* compared with controls, by an invasive protocol under anesthesia and by telemetric method during conscious, unrestrained status. Mice with a cardiac loss of *Prdm16* had higher heart-to-body weight ratios and upregulated atrial natriuretic peptide, suggesting cardiac hypertrophy. Plasma aldosterone-to-renin activity ratios and plasma sodium levels decreased in *Prdm16*-deficient mice versus control. By RNA-seq and in subsequent functional analyses, *Prdm16*-null hearts were enriched in factors that regulate BP, including *Adra1a*, *Nos1*, *Nppa*, and *Nppb*. The inhibition of nitric oxide synthase 1 (NOS1) reverted the decrease in BP in cardiac-specific *Prdm16* knockout mice. Mice with cardiac deficiency of *Prdm16* present with hypotension and cardiac hypertrophy. Further, our findings suggest that the increased expression of NOS1 causes hypotension in mice with a cardiac-null mutation of *Prdm16*. These results provide novel insights into the molecular mechanisms of hypotension in subjects with HF and contribute to our understanding of how hypotension develops in patients with HF.

## Introduction

Cardiovascular disease, comprising coronary heart disease, heart failure (HF), stroke, and hypertension, poses tremendous health and economic burdens globally; the prevalence of cardiovascular disease in adults was 49.2% in 2018, increasing with age in the United States [1]. Among cardiovascular diseases, HF is a leading cause of morbidity and mortality worldwide

**Funding:** This work was supported by the Basic Science Research Program and the Bio & Medical Technology Development Program, which are managed by the National Research Foundation of Korea (NRF) (2018R1D1A1B07042484, 2021R1I1A1A01049398, 2019R1A2C1083980, and 2019M3E5D3073365). The funders had no role in study design, data collection and analysis, decision to publish, or preparation of the manuscript. J.O.K., 2018R1D1A1B07042484, National Research Foundation of Korea, https://www.nrf.re.kr/biz/info/info/view?menu_no=378&biz_no=390, NO J.O.K., 2021R1I1A1A01049398, National Research Foundation of Korea, https://www.nrf.re.kr/biz/info/info/view?menu_no=378&biz_no=42, NO B.O., 2019R1A2C1083980, National Research Foundation of Korea, https://www.nrf.re.kr/biz/info/info/view?menu_no=378&biz_no=388, NO B.O., 2019M3E5D3073365, National Research Foundation of Korea, https://www.nrf.re.kr/biz/info/info/view?menu_no=378&biz_no=104, NO

**Competing interests:** The authors have declared that no competing interests exist.

[1]. Moreover, the rate of hypertension as a comorbidity in patients with established HF has been estimated to range from 25% to 70%, as reported by epidemiological studies in Europe [2,3]. Hypotension develops in 10% to 15% of patients with HF in clinical trials, and its incidence rises significantly in young women with incident HF [4–6].

Prdm16 is a transcription factor and chromatin modulator that regulates various processes, such as hematopoiesis [7], palatogenesis [8], neuronal specification [9], and adipogenesis [10]. Recently, the functions of Prdm16 in cardiac development and cardiovascular disease were described. A study of mice in which *Prdm16* was deleted in endothelial cells suggested that Prdm16 is required for the recovery of arterial flow under pathological challenge, based on its role in maintaining endothelial function [11].

We and others have reported results that implicate Prdm16 in certain types of cardiomyopathy, HF, and cardiac conduction abnormalities [12–14]. Arndt et al. (2013) noted that the loss of *PRDM16* causes left ventricular noncompaction (LVNC) and dilated cardiomyopathy (DCM) in patients with 1p36 deletion syndrome [12]. We have also shown that mice with cardiac deletion of *Prdm16* (*Prdm16*flox/flox; Myh6-Cre) experience dysfunctional cardiac conduction and cardiomyopathy-associated phenotypes, suggesting that impairments in ion homeostasis underlie these phenotypes [14]. Cibi et al. (2020) observed the development of hypertrophic cardiomyopathy in aged mice with a cardiac-null mutation of *Prdm16* (*Prdm16*flox/flox; Mesp1-Cre) [13], which experience progressive cardiac dysfunction by echocardiography, as evidenced by decreased left ventricular ejection fraction (LVEF); fractional shortening (FS); and increased end-systolic volume (ESV), end-diastolic volume (EDV), and left ventricle internal diameter during end-systole (LVIDes) [13]. Ultimately, cardiac-specific deletion of *Prdm16* leads to HF [13].

HF is subdivided into several types, based on LVEF: HF with preserved LVEF and HF with reduced LVEF [15]. HF with preserved LVEF is clinically defined as diastolic dysfunction—the inability of the ventricles to relax properly—whereas HF with reduced LVEF occurs when the ventricle loses its ability to contract normally [16,17]. Patients with HF with reduced LVEF have higher levels of circulating brain natriuretic peptide (BNP), a common biomarker for HF, and a higher mortality rate than those with HF with preserved LVEF [15].

Contrary to the well-established linear association between elevated blood pressure (BP) and cardiovascular events in the general population [18], observational studies have reported that decreasing systolic blood pressure (SBP) does not necessarily reduce cardiovascular mortality and hospitalization in patients with HF, particularly those with HF with reduced LVEF [3,6]. Although BP is an important factor in managing the mortality of HF, there are few studies that have examined changes in BP in cases with HF.

Thus, using adult mice (aged 5 months) that harbor cardiac-specific *Prdm16* conditional knockout alleles, we analyzed changes in BP to determine the underlying mechanisms of BP-associated changes in a murine model with HF or cardiomyopathy.

## Methods

### Animal research and ethics statement

*Prdm16*flox/flox homozygous (B6.129-*Prdm16*tm1.1Brsp/J) and Myh6-Cre transgenic mice (B6.FVB-Tg(Myh6-cre)2182Mds/J; αMyHC-Cre) have been described [19,20]. The *Myh6-Cre* (*αMyHC-Cre*) transgene directs the expression of Cre recombinase under the cardiac-specific alpha myosin-heavy chain (*Myh6*) promoter, which effects over 90% recombination in cardiomyocytes from the adult and embryonic mouse, starting from E8 [19].

*Prdm16*flox/flox homozygous mice were mated with Myh6-Cre mice to produce cardiac-specific *Prdm16* heterozygous mice (*Prdm16*flox/+; Myh6-Cre), which were then backcrossed with

*Prdm16*flox/flox animals to generate mice with cardiac-specific knockout of *Prdm16* (*Prdm16*flox/flox; Myh6-Cre), termed *Prdm16*CKO; *Prdm16* heterozygotes (*Prdm16*flox/+; Myh6-Cre) were designated *Prdm16*HET. Control mice (*Prdm16*+/+; Myh6-Cre) were generated by backcrossing heterozygous *Prdm16* mice (*Prdm16*flox/+; Myh6-Cre) with *Prdm16*flox/+ mice. Instead of *Prdm16*flox alleles without Myh6-Cre, no *Prdm16*flox alleles with Myh6-Cre (*Prdm16*+/+; Myh6-Cre) were used as a control, because cardiac function decreases from age 6 months of age due to Myh6-Cre [21].

All mice were housed and handled in a pathogen-free facility at College of Pharmacy, Kyung Hee University, per the Guide for the Care and Use of Laboratory Animals. The mice were maintained on a 12-hour light/dark cycle at constant temperature with free access to food (LabDiet 5L79, St. Louis, MO, USA) and water. Every effort was made to minimize the number of animals that was sacrificed and their suffering. Animals were anesthetized by intra-peritoneal (i.p.) injection of tribromoethanol (Avertin, 18 ml of working solution per kg body weight), the working solution of which was diluted to 25 μg/ml in 0.9% NaCl from stock solution (1 g/ml 2,2,2-tribromoethanol, dissolved in tertiary amyl alcohol), and euthanized by cervical dislocation. The experiments were approved by the local committee for the Care and Use of Laboratory Animals, Kyung Hee University (license number: KHSASP-19-328 and KHSASP-20-693).

## Genotyping of *Prdm16*flox and Cre alleles

Genomic DNA was extracted from mouse ears using the Thermo Scientific Phire Animal Tissue Direct PCR Kit (Waltham, MA, USA). The progeny of the cross between *Prdm16*flox heterozygotes or homozygotes and Myh6-Cre mice were genotyped with regard to the *Prdm16*flox and Cre alleles by PCR with the primer sets in S1 Table in S1 File.

## Invasive BP measurement

The invasive protocol for BP measurement has been described [22,23]. In brief, BP was recorded intra-arterially with a computerized data acquisition system (AD Instruments, Bella Vista, Australia). First, mice (age 18–20 weeks, > 20 g) were anesthetized by i.p. injection of tribromoethanol (Avertin, 18 ml/kg). The intra-arterial catheter (polyethylene tube; 0.2 mm I. D., 0.5 mm O.D.; Natsume, Tokyo, Japan), filled with 0.9% NaCl that contained 100 U/ml heparin, was connected to the system. Then, under anesthesia, a small incision was made to draw out the right carotid artery of the mouse. The catheter was inserted into the right carotid artery through a small puncture and ligated tightly with 4–0 silk suture thread. Next, BP in the vessel was transmitted along the catheter to the transducer's diaphragm (MLT0699 Disposable BP Transducer, AD Instruments, Bella Vista, Australia). The diaphragm signal was amplified through a bridge amplifier and recorded on a Power Lab system (Lab Chart 7.2, AD Instruments, Bella Vista, Australia).

BP was monitored for 2 hours after injection of the anesthetic and expressed as the average value from 40 to 80 min after injection of the anesthetic.

## Telemetric BP measurement

Mice (age 18–20 weeks, > 20 g) were anesthetized by i.p. injection of tribromoethanol (Avertin, 18 ml/kg). A telemetric device (HD-X10; Data Sciences International, St. Paul, MN, USA) was implanted as described [24,25]. In brief, mice were kept on a heating pad (37˚C) throughout implantation of the telemeter in the left common carotid artery. A skin incision was made below the neck, and the telemeter body was tunneled into the subcutaneous pocket and placed in the peritoneal cavity. The connected catheter was inserted into the carotid artery, and the

initial skin incision was sutured. Mice were maintained on a heating pad, administered an analgesic (1 mg/ml acetaminophen) in drinking water for 24 hr during the postsurgery period, and housed individually for 7 days to recover. BP was monitored via an implanted telemeter in conscious, unrestrained mice for 24 hours to obtain daytime and nighttime recordings.

## Biochemical analyses in plasma and urea

Blood was collected into lithium heparin capillary collection tubes (RAM Scientific) from anesthetized mice by puncture of the carotid artery. Blood samples were centrifuged at 5000 x g for 10 min at 4˚C, and the supernatant plasma was frozen and kept at -70˚C. For urine collection, mice were housed in an individual metabolic cage (Korea Institute of Science and Technology, Seoul, Korea) for 3 consecutive days. Urine was collected every 24 hours for the last 2 days and centrifuged at 20,000 x g for 5 min at 4˚C; the supernatant was frozen and stored at -70˚C. Plasma and urinary ion concentrations were determined with the Cobas c502 Module for Clinical Chemistry (Roche, Germany) at Seoul Medical Science Institute (Kyung Ki Do, Korea). Plasma urea nitrogen concentrations were determined by enzymatic method at Seoul Medical Science Institute (Kyung Ki Do, Korea). Plasma renin activity was measured using a fluorometric renin assay kit (Abcam, ab138875). Atrial natriuretic peptide (ANP) in plasma and cardiac tissues was measured using the Mouse ANP ELISA kit (Novus Biologicals, NBP2-66733, Colorado, USA). Plasma aldosterone was measured using the Aldosterone EIA kit (ENZO Life Sciences, ADI-900-173, Farmingdale, New York, USA).

## Quantitative real-time PCR

Total RNA was extracted from mouse left ventricles (aged 1 and 5 months, male or female) using TRIzol (Invitrogen, Carlsbad, CA, USA). cDNA was synthesized from 500 ng of total RNA using the PrimeScript™ RT kit (TaKaRa, Shiga, Japan) per the manufacturer. Quantitative real-time PCR was performed using SYBR Green I (TaKaRa, Shiga, Japan) on an ABI Step One Real-Time PCR system (Applied Biosystems, Foster, CA, USA) using the following program: 45 cycles at 95˚C for 10 s, 60˚C for 15 s, and 72˚C for 20 s; the primer sequences are listed in S2 Table in S1 File. The data were expressed as the average relative mRNA level in each group. Relative mRNA expression was calculated as follows: the Ct value of *Gapdh* was subtracted from that of a target gene, and delta Ct ($\Delta$Ct) was converted to the linear term $2^{-\Delta Ct}$.

## RNA sequencing and differential expression analysis

Based on prior RNA sequencing, the resulting differentially expressed genes (DEGs) have been reported [14]. Briefly, total RNA was isolated from mouse left ventricles (*Prdm16*$^{flox/flox}$; Myh6-Cre versus *Prdm16*$^{flox/flox}$, age 1 month, male) using TRIzol (Invitrogen, Carlsbad, CA, USA). RNA-seq libraries were prepared with the TruSeq Stranded mRNA LT Sample Prep Kit (Illumina, San Diego, CA, USA) per the manufacturer's instructions. Sequencing was performed on a NovaSeq 6000 System (Illumina) using the NovaSeq 6000 S4 Reagent Kit per the manufacturer's instructions. Reads were mapped to mm10 build of the mouse genome using the STAR-RSEM mapping program. Differential expression was analyzed using TPM (transcripts per million) values following quantile normalization. Of 24,532 genes, 15,660 genes with TPM values ($> 0$) were analyzed for differential expression, based on the $\log_2$ fold-change (*Prdm16*$^{flox/flox}$; Myh6-Cre versus *Prdm16*$^{flox/flox}$). Ultimately, 772 genes were selected as DEGs that changed by more than 1.6-fold (772 genes $> |1.6|$).

## Functional gene ontology and pathway analysis

Seven hundred seventy-two DEGs with a $\log_2$-fold change $> |1.6|$ were analyzed to identify functionally enriched processes with gene ontology (GO) using PANTHER14.1 [26]. All 3 PANTHER GO-Slim categories (Biological Process, Molecular Function, and Cellular Component) were included. The false discovery rate (FDR) *P*-value for the cutoff was 0.05, and the threshold for fold-enrichment was more than 2. Of 772 DEGs, 112 genes pertained to specific GO classes with significantly functional expression patterns. Of the GO classes, the term "Regulation of blood pressure" ranked highest in fold-enrichment (FDR *P*-value = 0.0274, fold-enrichment = 11.15), with only 4 genes (*Adra1a*, *Nos1*, *Nppa*, and *Nppb*) included. For validation, qRT-PCR of these 4 genes was performed in 1-month-old males and 5-month-old, female mouse left ventricular tissue (*Prdm16*[CKO], *Prdm16*[HET] versus control).

## Pharmacological tests of BP

The following drugs were tested: A61603 hydrobromide (agonist of α-1A adrenergic receptor [Adra1a]; 0.2 µg/kg body weight i.p. in saline; 2 mg/ml stock in distilled $H_2O$; Tocris Bioscience, Avonmouth, Bristol, United Kingdom) [27]; phenylephrine hydrochloride (PE; non-selective agonist of α1-adrenoceptor; 25 µg/kg body weight i.p. in saline; 50 mg/ml stock in distilled $H_2O$; SigmaAldrich, USA) [27]; L-NAME hydrochloride (NAME; non-selective inhibitor of nitric oxide synthase [NOS]; 100 mg/kg body weight i.p. in saline; 50 mg/ml stock in distilled $H_2O$; SigmaAldrich, USA) [28]; 7-nitroindazole (7-NI; selective inhibitor of NOS1; 10, 25 mg/kg body weight i.p. in saline; 81.57 mg/ml stock in dimethyl sulfoxide [DMSO]; Tocris Bioscience, Avonmouth, Bristol, United Kingdom) [28–30]; atropine (Atro; antagonist of muscarinic acetylcholine receptors; 1 mg/kg body weight i.p. in saline; 10 mg/ml stock in ethanol; SigmaAldrich, USA) [27]; isoprenaline (Iso; non-selective agonist of β-adrenergic receptors; 100 µg/kg body weight i.p. in saline; 100 mg/ml stock in distilled $H_2O$; SigmaAldrich, USA) [27]; and anantin (antagonist of natriuretic peptide receptor [NPR]-A; 100 µg/kg body weight i.p. in saline 3 times per day for 2 days; 1 mg/ml stock in 0.5% acetic acid; US Biological Life Sciences, Salem, MA, USA) [31,32].

BP was recorded via telemetric implants in conscious mice before and after drug treatment. Before drug injection, the baseline BP was established over 30–60 min, after which the drugs were administered i.p. (5 µl/g bodyweight); then, the preinfusion baseline was reestablished over 60–180 min. For A61603, PE, atropine, and isoprenaline, the mean of the 2-min period immediately before injection was used as the baseline, and then, the 20-s mean peak response over 10 minutes after injection was recorded as the treatment value. For L-NAME and 7-NI, the 20-min mean value before injection and the mean of the 20-min peak response for 60 minutes after injection were considered the baseline and treatment value, respectively. Anantin was injected 3 times per day for 2 days i.p., and the mean of the 6-h period after the last injection was regarded as the treatment value.

## Western blot

Total proteins were extracted from mouse left ventricles (5-month-old mice) using PRO-PREP (Intron Biotechnology, Gyeoggi-Do, Korea) per the manufacturer, and protein concentration was determined by Bradford assay [33]. Total proteins (10 to 40 µg) were separated by 8–10% SDS-PAGE and transferred to a nitrocellulose membrane (Pall, Ann Arbor, MI, USA). The membrane was blocked in 5% skim milk for 1 hour at room temperature and then incubated overnight at 4°C with primary antibody (S3 Table in S1 File). The blot was then washed and incubated with horseradish peroxidase-conjugated secondary antibody (Santa Cruz Biotechnology, Inc., Santa Cruz, CA, USA) for 1 hour at room temperature. Signals were detected

using Luminol (Santa Cruz Biotechnology) and exposed to x-ray films (Agfa-Health Care NV, Mortsel, Belgium). Relative band densities were determined using ImageJ (1.46). Actin was used as a loading control. Total proteins were normalized to actin, and phosphorylated proteins were normalized to total protein levels.

## Statistical analysis

The statistical analysis was performed using SPSS (IBM SPSS Statistics 25.0). The data were analyzed with the genotype and experimental conditions blinded. Two-group comparisons were performed using 2-tailed student's *t*-test. The numbers in each figure indicate the number of mice that was used for each group. The results are reported as mean ± SEM. Differences with $P < 0.05$ were considered to be statistically significant: a, $P < 0.05$; b, $P < 0.01$; and c, $P < 0.005$ versus respective controls.

## Results

### BP in mice with cardiac-specific null mutation of *Prdm16*

We measured BP in 5-month-old *Prdm16*^CKO mice using an invasive protocol under anesthesia and by telemetric method during conscious, unrestrained status. Mean BP under anesthesia decreased significantly only in female *Prdm16*^CKO versus control mice (88% of control; Fig 1A and S1A Fig in S1 File). Consistent with the invasive measurements, telemetric SBP and DBP in conscious mice fell significantly only in female *Prdm16*^CKO mice compared with controls during the day and night (91% to 92% of control; Fig 1B and S1B Fig in S1 File). Heart rate did not differ significantly between *Prdm16*^CKO and control mice (Fig 1B and S1B Fig in S1 File).

### Cardiac hypertrophy in mice with cardiac loss of *Prdm16*

Cibi et al. (2020) have reported the development of hypertrophic cardiomyopathy in 2 mouse strains with a cardiac-null mutation of *Prdm16* (*Prdm16*^flox/flox; Mesp1-Cre and *Prdm16*^flox/flox; αMHC^MerCreMer/+), resulting in HF [13]. We measured the weight of the hearts and also observed cardiac hypertrophy, based on the increases in heart weight and heart-to-body weight ratio in *Prdm16*^CKO compared with control mice (Fig 2A–2D and S2 Fig in S1 File). Atrial natriuretic peptide (ANP), an established hypertrophy marker, was upregulated in the cardiac ventricular tissues and plasma of *Prdm16*^CKO versus controls (Fig 2E and 2F), confirming the involvement of a loss in *Prdm16* in cardiac hypertrophy.

In determining the molecular mechanisms that correlate with the loss of *Prdm16*, we have previously reported RNAseq data in left ventricular tissue from 1-month-old, male *Prdm16*^CKO and wild-type mice [14]. Consistent with the RNAseq data from young *Prdm16*^flox/flox; Mesp1-Cre mice, we confirmed that many genes that are implicated in cardiac hypertrophy and oxidative stress were upregulated or downregulated in the cardiac tissues of 1-month-old, male *Prdm16*^CKO compared with control (S3 Fig in S1 File). To validate the RNAseq results, we performed the qRT-PCR in the left ventricular tissue of 5-month-old, female *Prdm16*^CKO and controls. We found that 18 of 24 genes were significantly increased in 5-month-old, female *Prdm16*^CKO compared with controls, supporting the implication of cardiac hypertrophy and oxidative stress in the underlying pathological changes upon the loss of *Prdm16* (S3 Fig in S1 File).

For example, the expression of natriuretic peptide A (*Nppa*), natriuretic peptide B (*Nppb*), myosin heavy chain 7 (*Myh7*), myosin light chain 4 (*Myl4*), myosin binding protein H-like (*Mybphl*), canopy FGF signaling regulator 2 (*Cnpy2*), cysteine and glycine-rich protein 3 (*Csrp3*), homeodomain-only protein X (*Hopx*), and peroxiredoxin 2 (*Prdx2*) was increased in

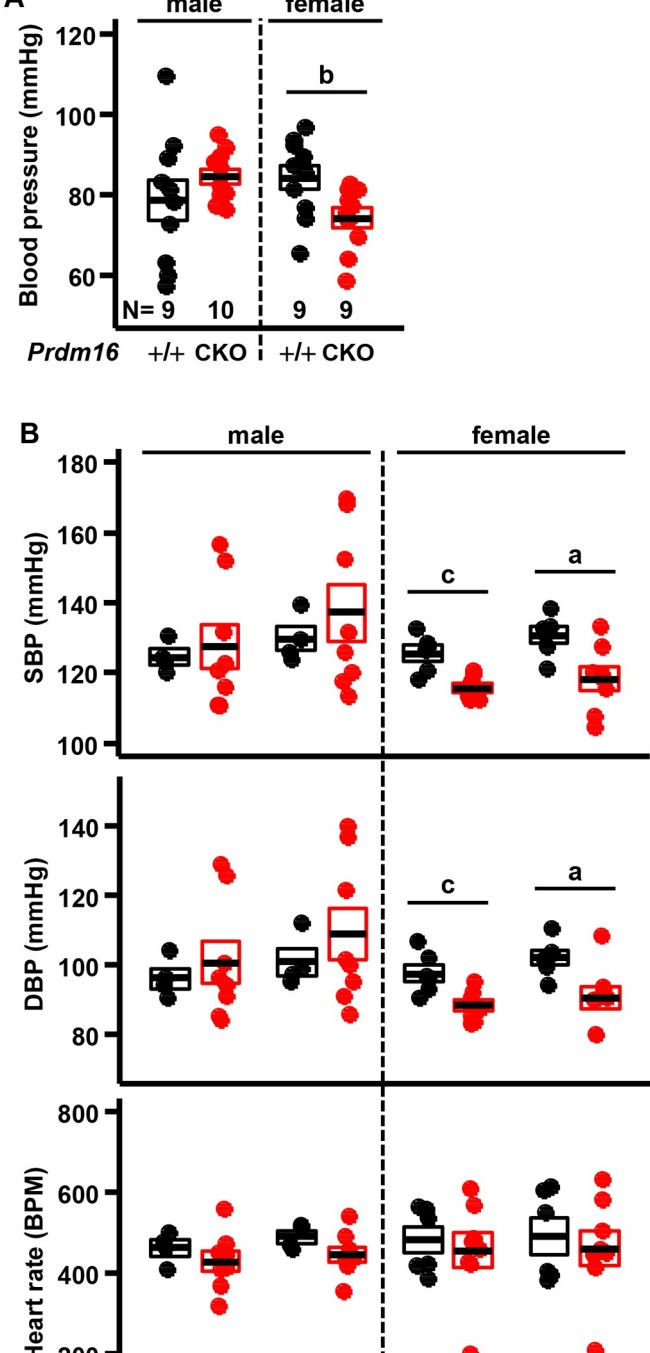

**Fig 1. Decreased BP in female mice with cardiac-specific inactivation of *Prdm16*.** (A) BP was measured intra-arterially under anesthesia separately in 5-month-old male and female control (+/+, *Prdm16*[+/+]; Myh6-Cre) and cardiac-specific knockout mice of *Prdm16* (CKO, *Prdm16*[flox/flox]; Myh6-Cre), termed *Prdm16*[CKO]. (B) SBP, DBP, and heart rate were measured via a telemetric implant for 24 hours (day and night) during conscious, unrestrained status 1 week after implantation. Error bars, mean ± SEM; number (N), number of mice; statistical significance by student's *t*-test, a ($P < 0.05$), b ($P < 0.01$), and c ($P < 0.005$).

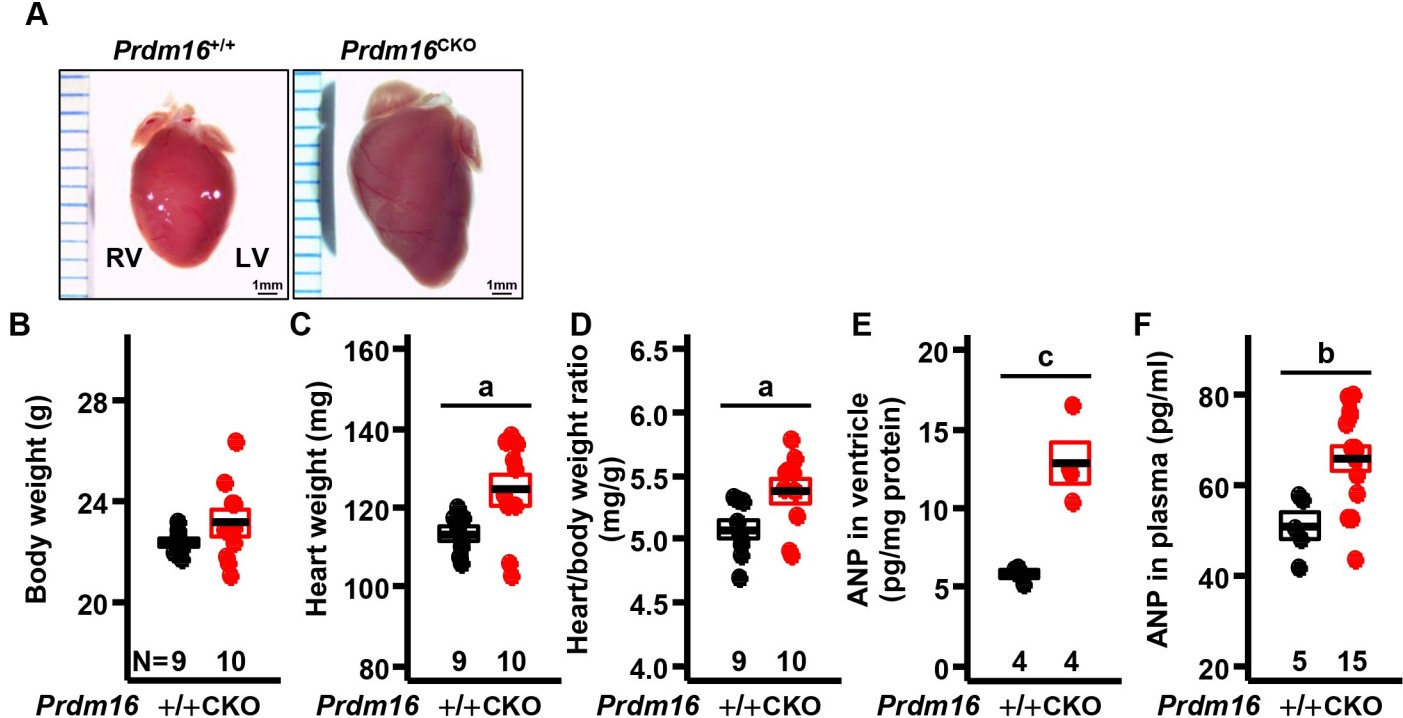

**Fig 2. Cardiac hypertrophy in mice with cardiac deletion of *Prdm16*.** (A) Shapes of whole hearts from 5-month-old female mice. (B) Body weight, (C) heart weight, and (D) heart-to-body weight ratio. (E) ANP levels in left ventricle and (F) plasma. Error bars, mean ± SEM; number (N), number of mice; statistical significance by student's *t*-test, a ($P < 0.05$), b ($P < 0.01$), and c ($P < 0.005$).

*Prdm16*<sup>CKO</sup> compared with control, suggesting that cardiac hypertrophy occurs in *Prdm16*<sup>CKO</sup> (S3 Fig in S1 File). In addition, many antioxidant genes, such as arachidonate 5-lipoxygenase (*Alox5*), gamma-glutamyl cyclotransferase (*Ggct*), glutathione peroxidase 1 (*Gpx1*), *Gpx3*, and glutathione S-transferase (*Gst*), were upregulated in *Prdm16*<sup>CKO</sup> hearts, suggesting increased oxidative stress (S3 Fig in S1 File).

### The renin-angiotensin-aldosterone system in cardiac-specific *Prdm16* knockout mice

Because the renin-angiotensin-aldosterone system (RAAS) is primarily responsible for regulating BP, we measured plasma aldosterone concentrations, plasma renin activity, and transcript levels of RAAS components. We found that plasma aldosterone levels and plasma renin activity did not change between female *Prdm16*<sup>CKO</sup> and controls, whereas the plasma aldosterone-to-renin activity ratio (ARR) decreased significantly (Fig 3A–3C), indicating hypotensive activity. However, in male mice, plasma aldosterone levels and plasma renin activity increased significantly in *Prdm16*<sup>CKO</sup> compared with controls, whereas the plasma aldosterone-to-renin activity ration (APR) did not change between them (S4 Fig in S1 File).

We also measured mRNA levels of RAAS components. In female *Prdm16*<sup>CKO</sup> animals, the transcripts of angiotensinogen (*Agt*) and angiotensin-converting enzyme (*Ace*) decreased in the liver and lung (systemic RAAS), whereas renin (*Ren1*) mRNA levels climbed in the kidney (intrarenal RAAS) versus control, implying differential compensatory responses between systemic and intrarenal RAAS activity (Fig 3D). Angiotensin type 1a receptor (*Agtr1a*), *Ace*, *Agtr2*, and *Ace2* transcripts were unchanged in the kidney between genotypes of female mice (Fig 3D).

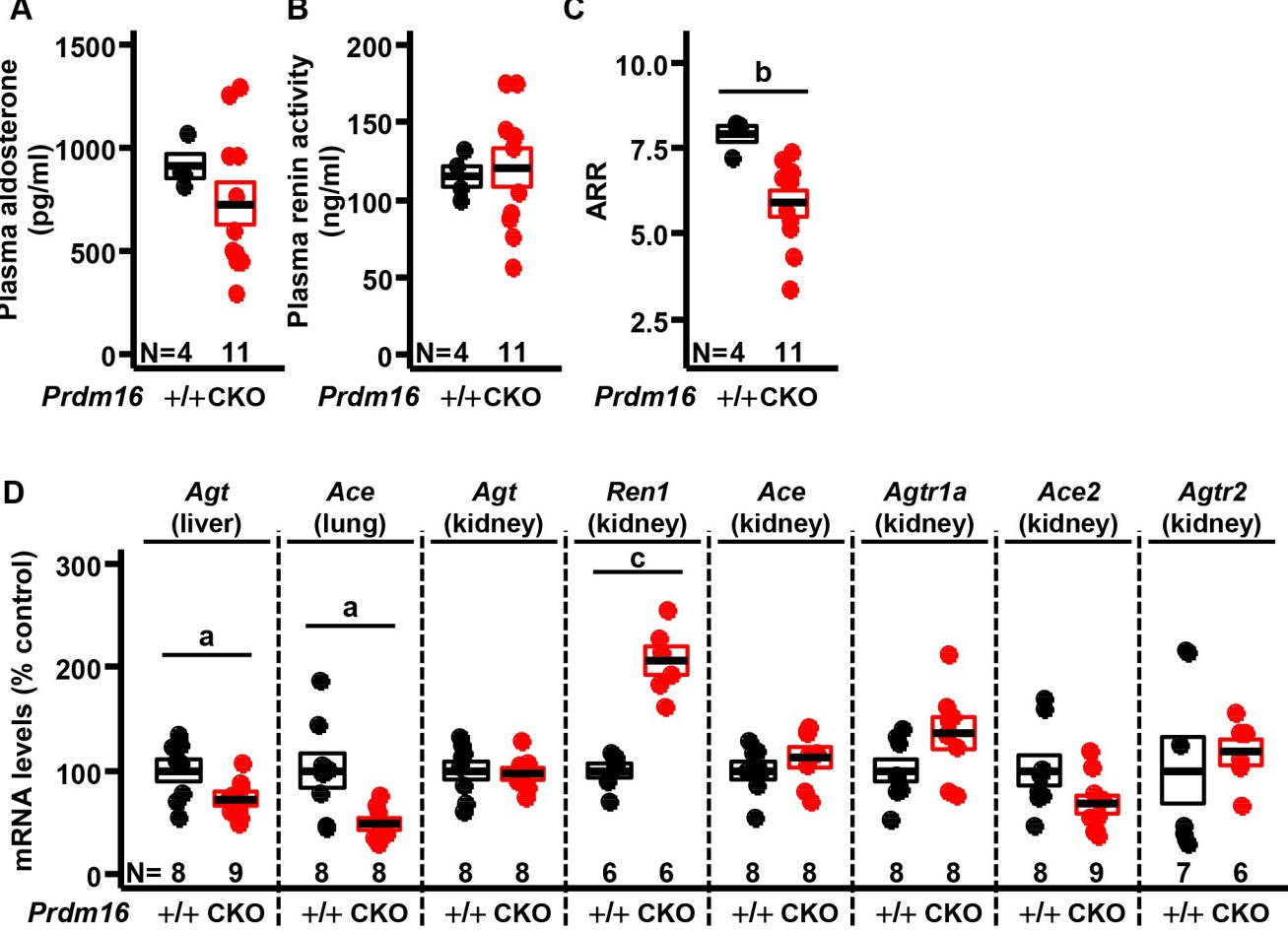

**Fig 3. Changes in the renin-angiotensin-aldosterone system in female mice with cardiac-specific null mutation of *Prdm16*.** (A) Aldosterone levels, (B) renin activity, and (C) aldosterone-to-renin activity ratio (ARR) in plasma. (D) mRNA levels in liver, lung, and kidney for components of the renin-angiotensin system. Error bars, mean ± SEM; number (N), number of mice; statistical significance by student's *t*-test, a ($P < 0.05$), b ($P < 0.01$), and c ($P < 0.005$).

We examined the depressor branch of RAAS pathways (ACE2/AT2R) in both male and female mice. Because sex hormones tend to favor the depressor branch of RAAS pathways in female [34], wherein estrogen increases the depressor pathway, such as ACE2/Ang1-7/AT2R, whereas testosterone amplifies the classical ACE/AngII/AT1R pressor pathway [35]. In our study, we measured the changes in transcriptional ratios of the depressor pathway, ACE2/AT2R, versus the pressor pathway, ACE/AT1R. The relative AT2R-to-AT1R ratio decreased significantly in kidney tissue of male *Prdm16*^CKO mice, but did not change in female *Prdm16*^CKO mice (S5 Fig in S1 File). This results imply that the depressor branch of RAAS pathways may play a partial role in the hypotensive activity in female *Prdm16*^CKO mice.

## Blood and urine biochemistry in mice with cardiac-null mutation of *Prdm16*

Body weight-normalized renal weight, creatinine clearance rate, and plasma urea nitrogen did not differ between *Prdm16*^CKO and control mice, irrespective of sex (Fig 4Ai–4Aiii and S2 and S6 Fig in S1 File), indicating that kidney function is normal in *Prdm16*^CKO mice. Plasma potassium,

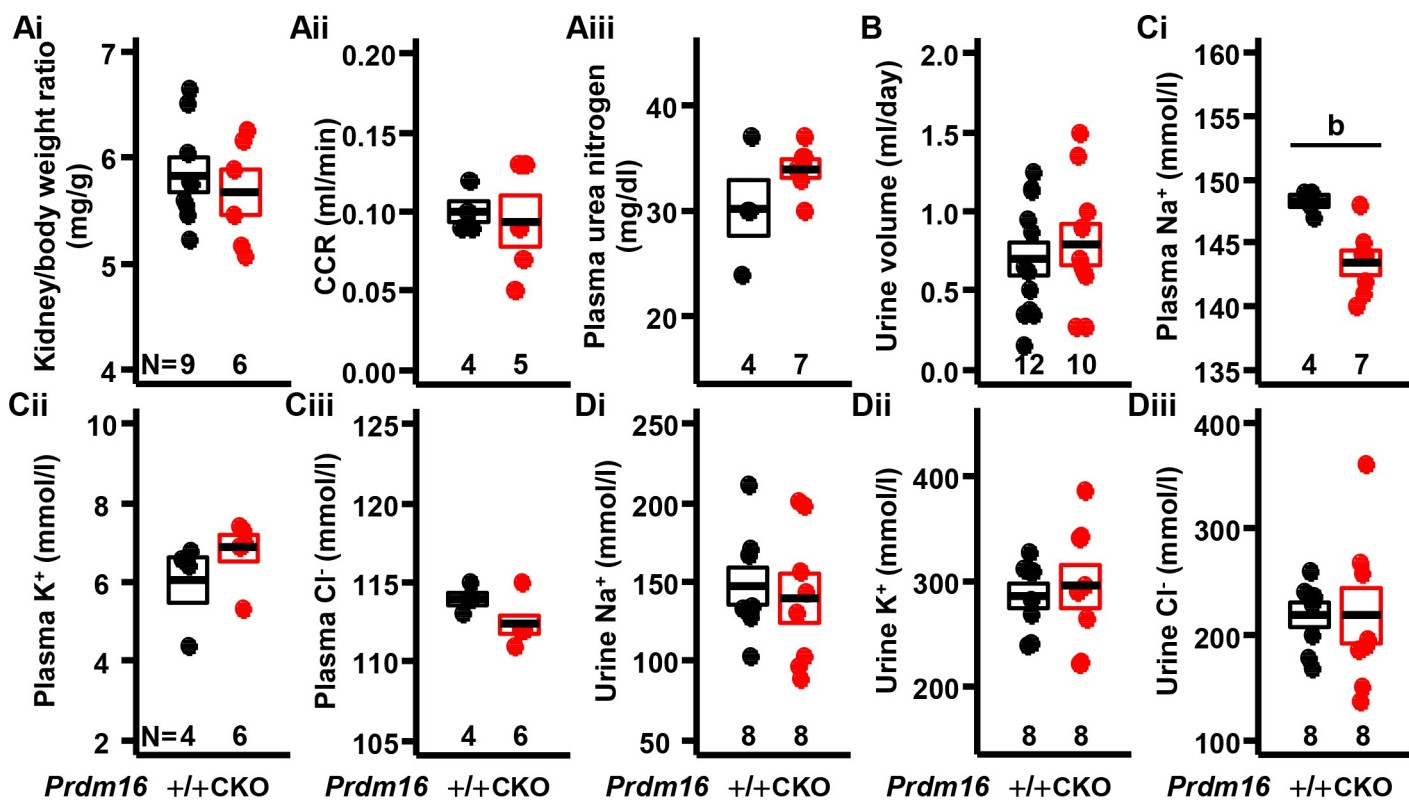

**Fig 4. Changes in electrolyte balance in cardiac-specific *Prdm16* knockout, female mice.** (Ai) Kidney-to-body weight ratio. (Aii) Creatinine clearance rate (CCR). (Aiii) Urea nitrogen in the plasma. (B) 24-hour urine volume. (Ci) Plasma sodium levels. (Cii) Plasma potassium levels. (Ciii) Plasma chloride levels. (Di) Urinary sodium levels. (Dii) Urinary potassium levels. (Diii) Urinary chloride levels. Error bars, mean ± SEM; number (N), number of mice; statistical significance by student's *t*-test, b ($P < 0.01$).

plasma chloride, urinary sodium, urinary potassium, and urinary chloride were also unchanged between *Prdm16*CKO and control mice (Fig 4Cii–4Ciii and 4Di–4Diii and S6 Fig in S1 File). Plasma sodium was lower in female *Prdm16*CKO versus control mice, but urinary sodium was lower only in male *Prdm16*CKO compared with the control, although urine volume was similar between groups (Fig 4B and 4Ci and S6 Fig in S1 File). These results imply that reduced sodium reabsorption contributes in part to the hypotensive activity in female *Prdm16*CKO mice.

## Transcriptional alterations in mouse left ventricular tissue on loss of *Prdm16*

As discussed, we have reported RNAseq data in left ventricular tissue from 1-month-old male *Prdm16*CKO and wild-type mice [14]. To identify target genes that correlated with the regulation of BP, the resulting 772 DEGs were reanalyzed with PANTHER GO-Slim, of which 112 were related to specific GO classes with significant expression patterns (FDR *P*-value < 0.05, fold-enrichment > 2.0; Fig 5A and S7 Fig in S1 File). The GO class "Regulation of blood pressure" consisted of 4 genes (*Adra1a*, *Nos1*, *Nppa*, and *Nppb*) that ranked highest in fold-enrichment (FDR *P*-value = 0.0274, fold enrichment = 11.15) and the expression patterns of which were comparable between the RNAseq and qRT-PCR data in 1-month-old male mice (Fig 5B). These transcriptional changes were validated by qRT-PCR in 5-month-old female mouse left ventricular tissue (*Prdm16*CKO versus control). We found that *Adra1a* transcripts decreased significantly in *Prdm16*CKO compared with the control, whereas *Nos1*, *Nppa*, and *Nppb* mRNA levels rose

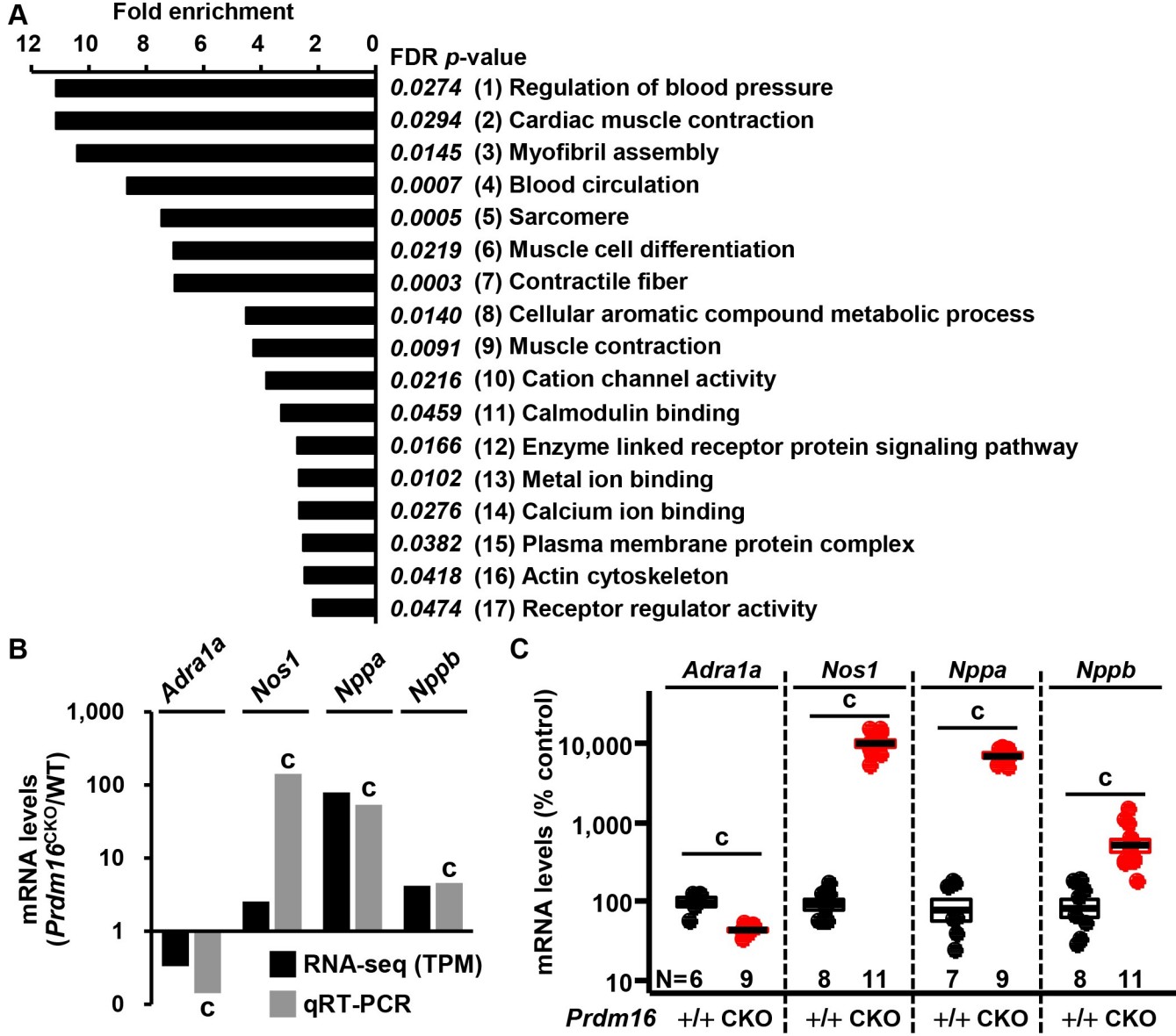

**Fig 5. Altered transcriptional profiles in mouse hearts with a *Prdm16* deletion and analysis of functional pathways.** RNA-seq analysis in 1-month-old male mouse left ventricular tissue, showing 772 differentially expressed genes (DEGs)—485 upregulated and 287 downregulated—in *Prdm16*^CKO versus wild-type hearts. (A) PANTHER gene ontology (GO) analysis was used for functional enrichment. Of 772 DEGs, 112 genes were categorized into 17 specific GO classes with significantly functional expression patterns (FDR P-value < 0.05, fold-enrichment > 2). The GO term "Regulation of blood pressure" ranked highest in fold-enrichment, with 4 genes (*Adra1a*, *Nos1*, *Nppa*, and *Nppb*) included. (B) Comparison of RNA-seq results and qRT-PCR for 4 genes in 1-month-old male mouse left ventricular tissue. (C) qRT-PCR of 4 genes in 5-month-old mouse left ventricular tissue in female *Prdm16*^CKO versus control. Error bars, mean ± SEM; number (N), number of mice; statistical significance by student's *t*-test, c (*P* < 0.005).

([Fig 5C] and S8 Fig in [S1 File]). These results suggest that alterations in the expression of α-1A adrenergic receptor, NOS1, ANP, and BNP underlie the variations in BP in *Prdm16*^CKO mice.

## Pharmacological interventions in mice with cardiac-null mutation of *Prdm16*

To determine the hypotensive mechanism in *Prdm16*^CKO mice, we analyzed telemetric BP recordings following pharmacological treatments that stimulate or inhibit the α-1 adrenergic system, β-adrenergic system, parasympathetic system, NOS, and ANP signaling.

A61603 (an α-1A adrenoceptor agonist; $EC_{50}$ = 0.3 µg/kg) significantly increased SBP in *Prdm16*CKO and control mice, with no difference between genotypes, at a dose (0.2 µg/kg) that is likely insufficient to fully activate less than the half-normal amount of α-1A adrenoceptor (Fig 6A and S9 Fig in S1 File) [27]. PE, a nonselective α-1 adrenoceptor agonist, also significantly increased SBP in *Prdm16*CKO and control mice at 25 µg/kg (Fig 6A and S9 Fig in S1 File). On treatment with A61603 or PE, heart rate rose insignificantly in *Prdm16*CKO versus control mice (Fig 6A). If hypotension in *Prdm16*CKO is attributable to the reduction in α-1A adrenoceptor, these doses of A61603 and PE are expected to increase BP in *Prdm16*CKO to a much less extent than control, but we saw no difference between genotypes [27]. These results suggest that the α-1A adrenergic receptor does not mediate the lower BP in *Prdm16*CKO mice.

Parasympathetic inhibition by atropine, a muscarinic receptor antagonist, significantly increased SBP and slightly increased heart rate in *Prdm16*CKO mice, whereas only heart rate rose significantly in control mice (Fig 6A and S9 Fig in S1 File) [27]. Final SBP did not differ between *Prdm16*CKO and control animals, but final heart rate differed significantly between genotypes (Fig 6A). This effect implies a relatively normal increase in basal sympathetic activity in both strains. β-adrenergic stimulation by isoprenaline lowered SBP and elevated heart rate significantly and similarly in *Prdm16*CKO and control mice (Fig 6A and S9 Fig in S1 File). Thus, β-adrenergic effects were unchanged in *Prdm16*CKO. Final SBP was significantly lower in *Prdm16*CKO than control mice, reflecting the disparate basal SBP between genotypes (Fig 6A). As evidenced on α-adrenergic and β-adrenergic stimulation by A61603, PE, and isoprenaline, mice that lack *Prdm16* are unlikely to suffer from deficiencies in the sympathetic system.

7-NI, a NOS1 inhibitor, significantly raised SBP in *Prdm16*CKO and control mice [28–30], as did L-NAME, a nonselective NOS inhibitor (Fig 6A)—to a greater extent in *Prdm16*CKO than control animals (Fig 6A and S9 Fig in S1 File) [28]. Neither inhibitor changed heart rate in either genotype (Fig 6A). These results suggest that increases in NOS1 are likely to cause hypotension in *Prdm16*CKO mice.

Anantin, an NPR-A antagonist, did not significantly alter SBP in *Prdm16*CKO or control mice, indicating that NPR-A is not involved in the decreased BP in *Prdm16*CKO mice (Fig 6A and S9 Fig in S1 File) [31,32].

To determine whether the increase in NOS expression was due to another constitutive cardiac NOS isoform, we measured NOS1 and NOS3 in left ventricular tissue by western blot. NOS1 but not NOS3 rose significantly in *Prdm16*CKO mouse heart tissue compared with control, suggesting that NOS1 is the major isoform (Fig 6B and 6C and S10A and S10B in S1 File). In kidney, *Nos2* and *Nos3* levels were higher in female, but not male, *Prdm16*CKO mice versus control (Fig 6D and S10C Fig in S1 File). These results indicate that L-NAME-dependent restoration of hypotension is attributed primarily to the inhibition of NOS1 in heart, although the contributions by renal NOS2 and NOS3 explain in part the greater increase on treatment of *Prdm16*CKO mice with L-NAME. In combination with the recovery results above, we suggest that NOS1 is chiefly responsible for hypotension in *Prdm16*CKO mice that present with cardiac hypertrophy.

## Discussion

Although BP is an important factor in the study of treatment strategies for HF, a molecular understanding of the changes in BP has not been clear in cases with HF, especially in a murine model. In this study, using mice with a cardiac-specific deficiency in *Prdm16*, we confirmed the development of cardiac hypertrophy and noted decreases in BP in *Prdm16*CKO mice, which we attributed to the upregulation of NOS1.

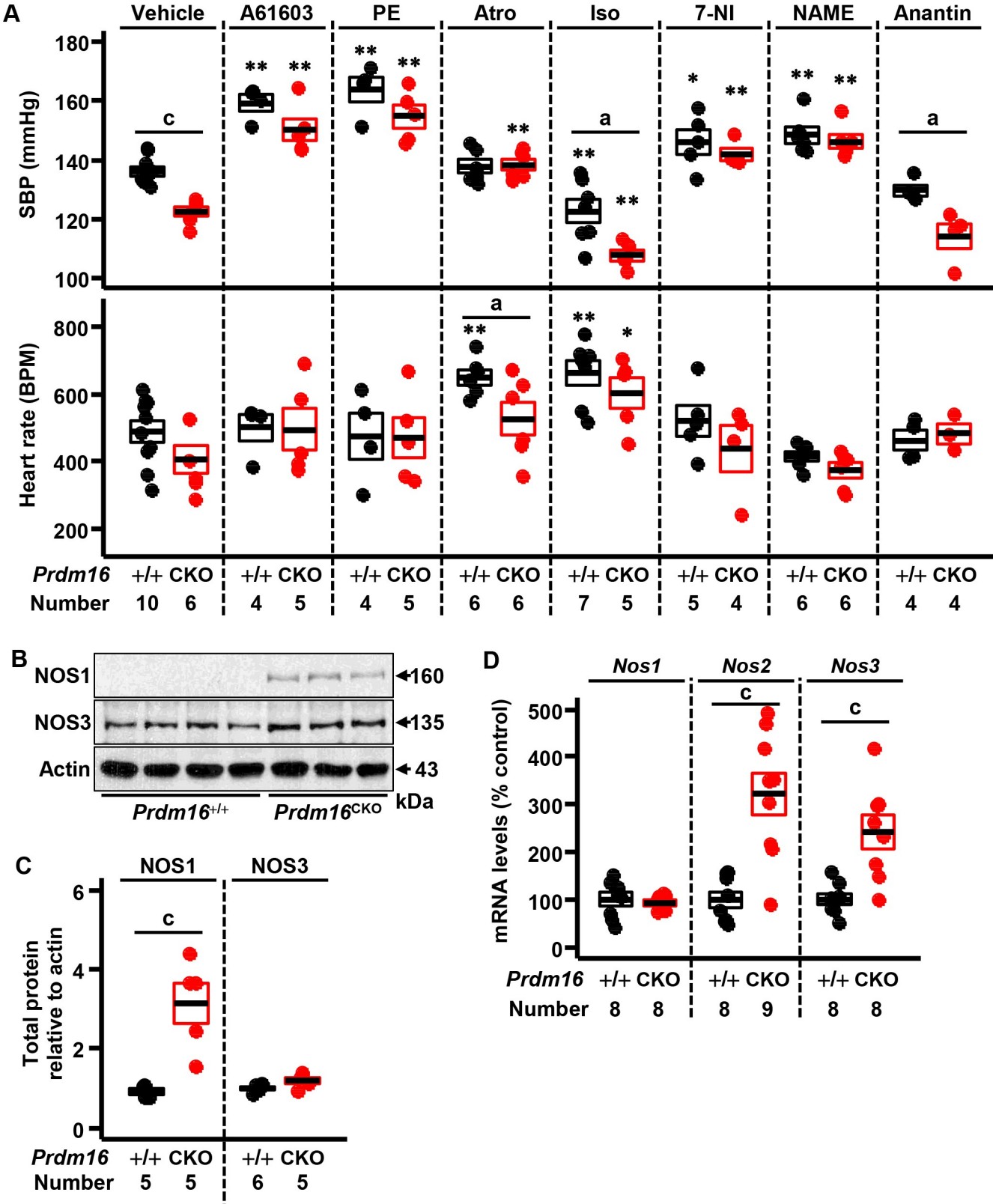

**Fig 6. Pharmacological restoration of hypotension in mice with cardiac deficiency of *Prdm16*.** (A) SBP and heart rate were measured via a telemetric implant in 5-month-old female *Prdm16*^CKO and control mice after vehicle (saline, except for atropine) or pharmacological administration. (B-C) NOS1 and

We have previously shown that the loss of *Prdm16* in mouse heart causes hypertrophic cardiomyopathy and a failure in cardiac conduction [14]. Similarly, Cibi et al. (2020) reported the development of hypertrophic cardiomyopathy in 2 strains with cardiac deficiency of *Prdm16*, noting HF with reduced LVEF and FS [13]. It appears that our strain (5-month-old *Prdm16*$^{flox/flox}$; Myh6-Cre) progresses to cardiac hypertrophy earlier than 9-month-old *Prdm16*$^{flox/flox}$; Mesp1-Cre mice, consistent with the other conditional strain (*Prdm16*$^{flox/flox}$; αMHC$^{MerCreMer/+}$) developing cardiac hypertrophy as early as age 3 months [13]. Despite this kinetic difference, the consistency in cardiac hypertrophy and expression of hypertrophy and oxidative stress genes in both strains suggests that our strain progresses to HF with reduced LVEF.

Although LVEF and BP are important independent predictors of death in patients with HF, reduced LVEF is a particularly strong risk factor in those with mean arterial pressure (MAP) or SBP $\leq$ 90 mm Hg but not in patients with MAP or SBP > 90 mm Hg [6,36]. Hypotension in HF patients with reduced LVEF can have several causes, such as low cardiac function, hypovolemia, treatment-related vasodilation, and altered vasoreactivity that is related to comorbidities, including diabetes [6,37]. Our cardiac-specific *Prdm16*-null mouse developed hypertrophic cardiomyopathy [14] and had low BP, strengthening it as a mouse model of HF and hypotension.

Our mice experienced an insignificant decrease in plasma aldosterone and a significant decline in ARR. ARR is strongly associated with BP [38], indicating a propensity in mice with a *Prdm16* deletion for hypotension. In addition, we noted less sodium reabsorption in mice with cardiac loss of *Prdm16*, also supporting the lowered BP.

A recent study reported strong evidence that NOS1/NO has important functions in the regulation of basal systemic vascular resistance and BP in healthy humans [39], in which 17 healthy normotensive men aged 24 years received acute intravenous infusions of a NOS1-selective inhibitor, S-methyl-$_L$-thiocitrulline (SMTC), or placebo on separate occasions. SMTC significantly increased systemic vascular resistance and DBP and lowered heart rate versus placebo [39]. This report and others suggest that NO influences systemic vascular resistance through local regulation of vascular tone, altering BP [39,40]. Moreover, the cardiovascular effects of inorganic nitrate supplementation have been studied, based on evidence that inorganic nitrate undergoes serial reductions to form NO and other closely related bioactive NO species [41]. Several groups have confirmed the BP-lowering effects of nitrate supplementation in healthy adults [42–44].

Studies in mice in which *Nos1* has been deleted suggest that NOS1-derived NO in the myocardium regulates basal myocardial inotropy and relaxation by inhibiting the sarcolemmal Ca$^{2+}$ current and promoting protein kinase A-dependent phosphorylation of phospholamban, independent of cGMP [45]. But, in AngII-treated left ventricular myocytes from hypertensive hearts or that conditionally overexpress *Nos1*, greater NOS1 activity facilitates myocyte relaxation through cGMP/protein kinase G (PKG)-dependent phosphorylation of downstream targets [46,47]. As a result, NOS1 reduces myofilament Ca$^{2+}$ sensitivity and promotes myocyte relaxation in hypertension [46,48].

In NO-sensitive guanylyl cyclase (GC) 1 or 2 knockout mice, NO-GC1 has been implicated as a target of NO to modulate renal blood flow and kidney function [49]. Animal studies suggest that the effect of NOS1-derived NO on renal renin release is also important in regulating BP [30,50]. In addition, studies in mice that lack *Nos3* or soluble guanylate cyclase α1 (*sGCα$_1$*) provide indirect evidence that the loss of NO-related signaling changes BP. Certain strains of NOS3-deficient or *sGCα$_1$*-deficient mice develop hypertension [22,51,52]. These studies indicate consistently that a reduction in NO/cGMP signaling increases the susceptibility to

hypertension in murine models. In contrast, similar to clinical trials, dietary supplementation with sodium nitrate in rats reduces BP—an effect that persists during the inhibition of NOS [53,54].

We did not determine which of these mechanisms contributes most to the hypotension in *Prdm16*$^{CKO}$ mice. However, the release of renal renin is unlikely to be responsible for the NOS1-associated changes in BP in these mice, because plasma renin activity did not change versus the control, and further, acute treatment with 7-NI does not affect renin secretion [55,56].

Based on previous reports and our findings, we propose that the induction of cardiac NOS1 in *Prdm16*$^{CKO}$ causes myocyte relaxation, likely through the modulation of NO-dependent targets in lowering BP in mice [46,47,56,57]. Further, the increase in renal NOSs/NO is likely to reinforce the hypotensive activity in *Prdm16*$^{CKO}$ mice.

The mechanisms that induce NOS1 in the diseased heart remain unknown. In the failing heart of humans and rats, NOS1 is upregulated and NOS3 decreases, in part through differential interactions with their allosteric regulators—caveolins—resulting in maintained myocardial NOS activity [58,59]. The transcription and translation of NOS1 are hypothesized to be oxygen-dependent, such as during hypoxia and in a milieu of reactive oxygen species [48]. A recent study by Cibi et al. (2020) has suggested that aged mouse hearts that are deficient for *Prdm16* have severe mitochondrial defects and constitutively express high levels of oxidative stress genes, such as *Gpx1*, *Gpx3*, *Alas1*, *Alas2*, and *Alox5*, compared with controls [13]. We also showed that many of these genes are upregulated in 1-month-old *Prdm16*$^{CKO}$ hearts and that the level of NOS1 transcripts is 142-fold higher in 1-month-old *Prdm16*$^{CKO}$ mice than in controls. Thus, it is likely that NOS1 is induced by high levels of reactive oxygen species in *Prdm16*$^{CKO}$ hearts. However, whether the initial induction of NOS1 is caused directly by transcriptional or post-transcriptional regulation by *Prdm16* or secondarily through reactive oxygen species in *Prdm16*$^{CKO}$ mice requires further study [13,14].

In this study, we focused on the molecular targets of changes in BP in a mouse model of HF and did not intend to study the effects on cardiomyopathy phenotypes. We administered drugs for short periods, except with anantin; thus, it is unlikely that any cardiomyopathy phenotype, including electrocardiography, was altered by any drug.

Although the sexual differences in BP changes were not of our major concern, we tried to speculate about why such BP changes have been observed only in female, but not in male, *Prdm16*$^{CKO}$ mice. As already described in the Result, we found that multiple results have supported a difference in BP changes between female and male *Prdm16*$^{CKO}$ mice, including the decrease in aldosterone-to-renin ratio and sodium reabsorption, and the stable activity of the depressor branch (AT2R) of RAAS pathways only in female *Prdm16*$^{CKO}$ mice. Based on previous reports and our findings, we can speculate that the induction of renal NOS2 and NOS3 as well as cardiac NOS1 contributes to the hypotensive activity in female, but not male, *Prdm16*$^{CKO}$ mice.

## Perspectives

HF is a leading cause of morbidity and mortality worldwide. In particular, hypertension or hypotension prevails as a comorbidity in patients with HF. Although BP is an important factor in managing the mortality of heart failure, the molecular mechanisms of changes in BP have not been clearly understood in cases of HF. We have demonstrated that mice with a cardiac-null mutation of *Prdm16* present with hypotension and cardiac hypertrophy. Further, our findings suggest that NOS1 is important in reducing BP in mice that present with cardiac hypertrophy due to a cardiac-specific deficiency in *Prdm16*. These results provide novel

insights into the molecular mechanism of hypotension in subjects with HF and increase our understanding of how hypotension develops in them.

## Supporting information

**S1 File.**
(DOCX)

**S1 Raw images. Original images for western blots.**
(PDF)

## Author Contributions

**Conceptualization:** Ji-One Kang.

**Data curation:** Ji-One Kang.

**Formal analysis:** Ji-One Kang.

**Funding acquisition:** Ji-One Kang, Bermseok Oh.

**Investigation:** Ji-One Kang, Tae Woong Ha.

**Methodology:** Ji-One Kang, Tae Woong Ha, Hae-Un Jung, Ji Eun Lim.

**Project administration:** Ji-One Kang.

**Resources:** Ji-One Kang, Tae Woong Ha.

**Supervision:** Ji-One Kang.

**Validation:** Ji Eun Lim.

**Writing – original draft:** Ji-One Kang.

**Writing – review & editing:** Ji-One Kang, Bermseok Oh.

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
