## [Decision Letter · Decision Letter 0]

27 Jan 2022

PONE-D-21-38135A cardiac-null mutation of Prdm16 causes hypotension in mice with cardiac hypertrophy via increased nitric oxide synthase 1PLOS ONE

Dear Dr. Kang,

Thank you for submitting your manuscript to PLOS ONE. After careful consideration, we feel that it has merit but does not fully meet PLOS ONE’s publication criteria as it currently stands. Therefore, we invite you to submit a revised version of the manuscript that addresses the points raised during the review process.

We look forward to receiving your revised manuscript.

Kind regards,

Paula A. da Costa Martins, Ph.D.

Academic Editor

PLOS ONE

Journal Requirements:

Reviewers' comments:

Reviewer's Responses to Questions

**Comments to the Author**

1. Is the manuscript technically sound, and do the data support the conclusions?

Reviewer #1: Yes

Reviewer #2: Yes

2. Has the statistical analysis been performed appropriately and rigorously? 

Reviewer #1: Yes

Reviewer #2: Yes

3. Have the authors made all data underlying the findings in their manuscript fully available?

Reviewer #1: Yes

Reviewer #2: No

4. Is the manuscript presented in an intelligible fashion and written in standard English?

Reviewer #1: Yes

Reviewer #2: Yes

5. Review Comments to the Author

Reviewer #1: Ji-One Kang et al. described the correlation between blood pressure and heart failure in cardiac-specific Prdm16-deficient mice. Not only the authors showed that female resulted more prone to develop hypotension and cardiac hypertrophy, but also they propose that NOS1 increased expression might cause the pathogenic phenotype. The study, however, lacks the mechanistic investigation underlining this condition.

Major comments

1. On lines 266-285, the authors describe the hypertrophic phenotype in Prdm16 null mice and they report data from RNA-seq performed on 1 m.o. mouse heterozygous for Prdm16 loss. This latter model doesn’t match the one used in the study for age (1 m.o. vs 5 m.o.), sex (male vs female), and genotype (heterozygous vs homozygous). qPCR on cardiac samples of 5 m.o. female Prdm16 CKO mice should be performed to validate in the proper model the RNA-seq findings.

2. In the discussion, the authors should speculate about why these findings have been observed only in female, and not male, mice

Minor comments

1. In figure legends, please add the explanation of f/f

2. In Fig. S2 please report the unit of heart/ and kidney/body weight ratios

3. Fig 4Ai is a partial repetition of Fig S2

Reviewer #2: The authors report on the axis Prdm16 and NOS1 and its role in hypotension and HF. The study is sound and the experimental setting logical and well performed There are however some points that deserve more attention:

- it is not clear whether the RNA seq data is available in any kind of repository and this available to the scientific community.

- regarding the RNA seq data in figure 5, why not showing a heatmap with most up and downregulated genes that are relevant for this manuscript? I think this would be more explanatory and also provide additional information.

- a more complete characterization of the cardiac phenotypes is necessary: histology (H&E, fibrosis, cell size), qunatification of cell size, capillary rarefaction, cross sectional stainings to see whether there is any changes in the ventricle wall, etc

- related to the above point in figure 2 heart weight to liver weight or to tibia length should be added.

- Figure 4: give symbols to mark statistical significant instead of letters

- all the mice used in the study are Myh6-Cre, there is no need to include this in all the figures. Just remove this and the figures will become more straightforward.

6. PLOS authors have the option to publish the peer review history of their article (what does this mean?). If published, this will include your full peer review and any attached files.

Reviewer #1: No

Reviewer #2: No

---

## [Author Response · Author response to Decision Letter 0]

17 Mar 2022

We greatly appreciate your (Editor’s and Reviewers’) time and efforts in reviewing our manuscript. We have tried to incorporate all of reviewer’s comments in the revised manuscript. Please find the highlighted changes in the Revised Manuscript and we provide a point-by-point response to each comment as below.

We uploaded our response to reviewers comments as an attachment. Please find it as a file.

---

## [Decision Letter · Decision Letter 1]

20 Apr 2022

A cardiac-null mutation of Prdm16 causes hypotension in mice with cardiac hypertrophy via increased nitric oxide synthase 1

PONE-D-21-38135R1

Dear Dr. Kang,

We’re pleased to inform you that your manuscript has been judged scientifically suitable for publication and will be formally accepted for publication once it meets all outstanding technical requirements.

Kind regards,

Paula A. da Costa Martins, Ph.D.

Academic Editor

PLOS ONE

Reviewers' comments:

Reviewer's Responses to Questions

**Comments to the Author**

1. If the authors have adequately addressed your comments raised in a previous round of review and you feel that this manuscript is now acceptable for publication, you may indicate that here to bypass the “Comments to the Author” section, enter your conflict of interest statement in the “Confidential to Editor” section, and submit your "Accept" recommendation.

Reviewer #1: All comments have been addressed

Reviewer #2: All comments have been addressed

2. Is the manuscript technically sound, and do the data support the conclusions?

Reviewer #1: Yes

Reviewer #2: Yes

3. Has the statistical analysis been performed appropriately and rigorously? 

Reviewer #1: Yes

Reviewer #2: Yes

4. Have the authors made all data underlying the findings in their manuscript fully available?

Reviewer #1: Yes

Reviewer #2: Yes

5. Is the manuscript presented in an intelligible fashion and written in standard English?

Reviewer #1: Yes

Reviewer #2: Yes

6. Review Comments to the Author

Reviewer #1: Dear Authors,

thank you for addressing all my comments; I think that the paper now is more complete. I have no other comments.

Reviewer #2: The authors have addressed all my comments in a satisfactory manner. I am only not sure whether all the sequencing data is available since it has been published before and some sets re-analysed in this manuscript

7. PLOS authors have the option to publish the peer review history of their article (what does this mean?). If published, this will include your full peer review and any attached files.

Reviewer #1: No

Reviewer #2: No

---

## [Editor Report · Acceptance letter]

12 Jul 2022

PONE-D-21-38135R1 

A cardiac-null mutation of *Prdm16* causes hypotension in mice with cardiac hypertrophy via increased nitric oxide synthase 1 

Dear Dr. Kang:

I'm pleased to inform you that your manuscript has been deemed suitable for publication in PLOS ONE. Congratulations! Your manuscript is now with our production department. 

Kind regards, 

on behalf of

Dr. Paula A. da Costa Martins 

Academic Editor

PLOS ONE